# SEAL: Simultaneous Label Hierarchy Exploration And Learning

## Abstract

Label hierarchy is an important source of external knowledge that can enhance classification performance. However, most existing methods rely on predefined label hierarchies that may not match the data distribution. To address this issue, we propose **S**imultaneous label hierarchy **E**xploration **A**nd **L**earning (SEAL), a new framework that explores the label hierarchy by augmenting the observed labels with latent labels that follow a prior hierarchical structure. Our approach uses a 1-Wasserstein metric over the tree metric space as an objective function, which enables us to simultaneously learn a data-driven label hierarchy and perform (semi-)supervised learning. We evaluate our method on several datasets and show that it achieves superior results in both supervised and semi-supervised scenarios.

## 1 Introduction

Labels play a crucial role in machine learning. They provide the supervision signal for learning models from annotated data. However, obtaining label annotations is often costly and time-consuming, which motivates the study of semi-supervised learning that leverages both labeled and unlabeled data (Chen et al., 2020; Assran et al., 2021; Wang et al., 2021a). A common technique for semi-supervised learning is also related to the label, specifically, using (pseudo-)labels (Berthelot et al., 2019b; Sohn et al., 2020; Gong et al., 2021). Unlabeled data is augmented in different ways (Xie et al., 2020), and pseudo labels are then generated from model predictions for different augmentations of the same data. The model is updated by enforcing the consistency of pseudo labels across augmentations. This technique is known as "consistency regularization" (Rasmus et al., 2015b).

Labels are also important for understanding data, as they link real-world observations with abstract semantics. It has been shown that exploiting hierarchical structures of label semantics can enhance the performance of supervised and semi-supervised learning. These structures can be obtained from external sources such as decision trees (Wan et al., 2020) and knowledge graphs (Miller, 1998; Speer et al., 2017). Once the label hierarchy is available, models can be trained by either (1) predicting hierarchical semantic embeddings jointly with labels (Deng et al., 2010; 2012; Barz & Denzler, 2019; Liu et al., 2020; Wang et al., 2021b; Karthik et al., 2021; Nassar et al., 2021) or (2) optimizing hierarchical objective functions that incorporate label relations (Bertinetto et al., 2020; Bilal et al., 2017; Garnot & Landrieu, 2020; Garg et al., 2022b). Alternatively, the structure can also be incorporated into the model architecture itself (Kontschieder et al., 2015; Wu et al., 2016; Wan et al., 2020; Chang et al., 2021; Garg et al., 2022a).

Although predefined label hierarchies are frequently used in the existing literature, they cannot always match the actual data distribution. However, not much effort has been made to derive the label hierarchy from a data-driven perspective. To address this issue, we propose **S**imultaneous label hierarchy **E**xploration **A**nd **L**earning (SEAL), which achieves two goals by incorporating an additional regularization term.

The first goal of SEAL is to identify the data-driven label hierarchy. This goal differs from hierarchical clustering, which discovers hierarchical structures from data that do not align with labels. SEAL expands the label alphabet by adding unobserved *latent* labels to the *observed* label alphabet. The data-driven label hierarchy is modeled by combining the predefined hierarchical structure of *latent* labels with the optimizable assignment between observed and latent labels. The observation

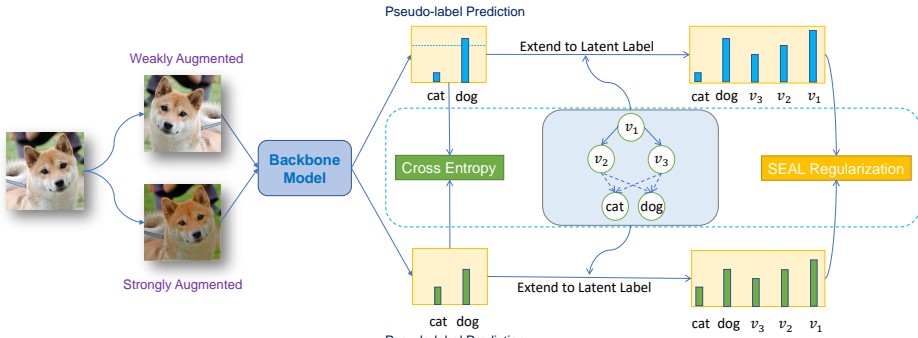

Figure 1: An illustration of the SEAL framework. After getting the predicted probability of a weakly augmented image on *observed* labels, the model shall use any pseudo-labeling techniques to give it a pseudo-label. Then the cross entropy loss between the pseudo label and predicted probability on the strong-augmented image will comprise part of the SEAL regularization. Given a (soft) tree hierarchy can introduce the *latent* labels. Extending the predicted probability on strong-augmented image and pseudo label to *total* labels will give a SEAL regularization loss which is another part of the SEAL regularization. Note that the backbone model and the SEAL regularization model (label hierarchies) can be updated via gradient descent simultaneously.

that inspires this approach is that the labels are typically a subset of concepts in a larger knowledge graph (Miller, 1998).

The second goal of SEAL is to improve the classification performance of state-of-the-art methods. To accomplish this goal, we propose a new regularization term that uses the model predictions and the label annotations on the *observed* label alphabet to encourage agreement on both *observed* and *latent* label alphabets. The confidence on *latent* labels is estimated by simulating the Markov chain based on the label hierarchy between *observed* and *latent* labels. This regularization term can be added to all existing approaches because of the universality of comparing the prediction and labels on the *observed* label alphabet. A diagram of SEAL is shown in Figure 1.

SEAL's soundness and effectiveness are validated theoretically and empirically. The regularization term can be interpreted as a relaxation of the tree Wasserstein metric (Le et al., 2019), and it can be used for optimization. Empirical evaluation demonstrates that adding the SEAL framework consistently and significantly improves the classification performance on supervised learning and various semi-supervised learning methods (Sohn et al., 2020; Zhang et al., 2021; Wang et al., 2022). SEAL also achieved a new state-of-the-art for semi-supervised learning on standard datasets. Additionally, our case study shows that the alignment between *observed* and *latent* labels also yields a meaningful label hierarchy.

## 2 BACKGROUND

In this section, we introduce the notations used throughout this paper first and then define the Tree-Wasserstein distance, which we use to define label hierarchy metrics in Section 3.

### 2.1 NOTATIONS

We consider supervised learning, where $D = \{(\boldsymbol{x}_i, y_i)\}_{i=1}^n$ is the labeled training set. Here, $x_i$ is an image, and $y_i$ is the corresponding (class) label in a set $\mathcal{O}$, where $|\mathcal{O}| = K$. Without further justification, $y$ is the categorical label, and $\delta_y$ is the one-hot label. Our goal is to learn a backbone model $f_\theta$ parameterized by $\theta$, which maps each image $\boldsymbol{x}_i$ to a probability over $\mathcal{O}$. We denote the predicted probability vector of image $\boldsymbol{x}$ as $f_\theta(\boldsymbol{x})$ and define the pseudo label of $\boldsymbol{x}$ as $\hat{y} = \arg\max_j f(\boldsymbol{x})^T e_j = \arg\max_j \Pr(j|\boldsymbol{x})$, where $e_j$ is the coordinate vector. The objective of supervised learning is to make the ground truth label $y$ and pseudo label $\hat{y}$ consistent.

We also consider semi-supervised learning, where an additional unlabeled dataset $D^u$ is provided. During training, we define the relative ratio $\mu$ as the number of unlabeled data to labeled data in a mini-batch. Following RandAugment (Cubuk et al., 2020), we shall define weak augmentation $P_{\text{wAug}}(\boldsymbol{x})$ and strong augmentation $P_{\text{sAug}}(\boldsymbol{x})$. These two augmentations are probability distributions of the views of $\boldsymbol{x}$ derived by argumentation while keeping the same pseudo label, the term weak and strong describes the distortion density. Moreover, $\boldsymbol{x}'$ and $\boldsymbol{x}''$ denote the argument data and $y'$ denotes the pseudo-label.

## 2.2 TREE-WASSERSTEIN DISTANCE

The Tree-Wasserstein distance (Le et al., 2019) is a 1-Wasserstein metric (Peyré et al., 2019) on a tree metric space $(\mathcal{X}, E, w)$, where $\mathcal{X}$ is the node set of a directed rooted tree, $E$ is the (weighted) edge set, and $w = (w_e)_{e \in E}$ denotes the weights of edges $e \in E$. The tree metric $d_\mathcal{X}$ between any two nodes of a tree is defined as the length of the shortest path between them. Given node $v \in \mathcal{X}$, let $\Gamma(v)$ be the set of nodes in the subtree of $\mathcal{X}$ whose root node is $v$. For each weighted edge $e \in E$, we denote the deeper (away from the root) level endpoints of weighted edge $e$ as $v_e$. Then, the Tree-Wasserstein metric can be computed in closed form, as shown in Theorem 1.

**Theorem 1 (Tree-Wasserstein distance (Le et al., 2019))** *Given two probability measures $\mu, \nu$ supported on a directed rooted tree $\mathcal{X}$, and choosing the ground metric as tree metric $d_\mathcal{X}$, then the Wasserstein-1 distance under Definition 4 can be reformulated as follows:*

$$W_{d_\mathcal{X}}(\mu, \nu) = \sum_{e \in E} w_e \left| \mu\left(\Gamma\left(v_e\right)\right) - \nu\left(\Gamma\left(v_e\right)\right) \right|. \tag{1}$$

## 3 THE SEAL FRAMEWORK

We propose a novel framework called Simultaneous label hierarchy Exploration and Learning (SEAL), which is motivated by probabilistic models that leverage latent structures to improve classification performance. In the SEAL framework, we introduce *latent* labels that are aligned with *observed* labels to capture the data-driven label hierarchy.

### 3.1 WHY LATENT STRUCTURE?

**Example 1** *Let us consider a classifier that predicts the probability of an image belonging whether an image belongs to the "apple" or "paint" class. Although the two classes may seem unrelated, they may share some hidden structures, such as the colors "red" and "green". Knowing the conditional probabilities of colors given each class, we can calculate the probability of an image being red given its probability of belonging to the "apple" class. Suppose we know the conditional probabilities $\Pr(red|apple) = 0.9$, $\Pr(green|apple) = 0.1$, $\Pr(red|paint) = 0.5$, and $\Pr(green|paint) = 0.5$. In this scenario, if the classifier assigns a probability of $0.8$ for an image being an apple, the question arises as to what is the probability that this image is red. By applying the law of probability, the answer is $0.8 \times 0.9 + 0.2 \times 0.5 = 0.82$. For a pictorial understanding, please see Figure 3 in the Appendix.*

### 3.2 LATENT STRUCTURE AND SEAL REGULARIZATION

The Example 1 illustrates that considering latent variables, such as color in this case, can provide more information to the classifier, leading to better performance. In the context of image classification, these latent variables can represent various factors, such as textures, shapes, and semantic meanings. However, identifying these latent variables and modeling their relationships with observed variables is not always straightforward, which is the focus of the SEAL framework.

Then we formally present the following definitions below. The set of labels $\mathcal{O}$ in the dataset is denoted as *observed* label alphabets. The set of latent labels $\mathcal{L}$ is denoted as *latent* label alphabet. We call $\mathcal{O} \cup \mathcal{L}$ *total* label alphabet. Let $|\mathcal{O}| = K$ and $|\mathcal{O} \cup \mathcal{L}| = N$ be the sizes of observed and total label alphabets. The relationship between *observed* and *latent* labels are described by (directed) graphs. Specifically, let $\mathbf{A}_1$ be the adjacency matrix for latent labels in $\mathcal{L}$ and $\mathbf{A}_2$ be the connection

matrix between $\mathcal{L}$ and $\mathcal{O}$. The adjacency matrix $\mathbf{A}$ for $\mathcal{L} \cup \mathcal{O}$ characterizes the latent structure.

$$\mathbf{A} = \begin{pmatrix} \mathbf{A}_1 & \mathbf{A}_2 \\ 0 & 0 \end{pmatrix}. \tag{2}$$

It is assumed that no connection exists inside $\mathcal{O}$. SEAL is targeted to discover the hierarchical structure of labels, therefore, additional assumptions are imposed on matrix $\mathbf{A}$. The key assumption of $\mathbf{A}$ follows the following theorem.

**Theorem 2 ((Takezawa et al., 2021))** *Suppose a directed graph $G$ having a total of $N$ nodes, which we denote as $\{v_1, v_2, ..., v_N\}$. If the adjacency matrix $\mathbf{A} \in \{0,1\}^{N \times N}$ of this graph satisfies the following conditions:*

*1. $\mathbf{A}$ is a strictly upper triangular matrix.*

*2. $\mathbf{A}^T \mathbf{1}_N = (0, 1, \cdots, 1)^\top$.*

*then $G$ is a directed rooted tree with $v_1$ as the root and $\mathbf{1}_N$ is an all-one vector.*

The hierarchical structure of observed labels is then described by the graph defined by $\mathbf{A}$, satisfying conditions in Theorem 2.

We introduce a weight matrix $\alpha = [\alpha_{sr}]_{s,r} \in \mathbb{R}^{(N-K) \times K}$ that describes the connection from $s \in \mathcal{L}$ to $r \in \mathcal{O}$ to further quantify how much an observed label contributes to a latent label. A **SEAL extension** is then defined by a five-tuple $(\mathcal{O}, \mathcal{L}, \mathbf{A}_1, \mathbf{A}_2, \alpha)$. We note that $\alpha$ should be related to the $\mathbf{A}_2$ and its specific formulation will be detailed in the following parts. Then, we are able to extend the model's prediction on the observed label alphabet $\mathcal{O}$ to the total label alphabet.

**Definition 1 (Total prediction and total target)** *Let $p_r$ be the probability of the label $r \in \mathcal{O}$, vector q on total label alphabet $\mathcal{O} \cup \mathcal{L}$ is*

$$[q(\mu)]_s = \begin{cases} \mu_s & \text{if } s \in \mathcal{O} \\ \sum_{r \in \mathcal{O}} \alpha_{sr} \mu_r & \text{if } s \in \mathcal{L} \end{cases}, \tag{3}$$

*where $s \in \mathcal{O} \cup \mathcal{L}$ and $\mu$ a probability distribution on $\mathcal{O} \cup \mathcal{L}$. Given a sample of input and (pseudo-)label $(\boldsymbol{x}, y)$, we note that the $p_r$ could be derived by both the model prediction $f_\theta(\boldsymbol{x})$, the one-hot label $\delta_y$, or the pseudo-label $\delta_{y'}$. Moreover, $q(f_\theta(\boldsymbol{x}))$ is the total prediction while $q(\delta_y)$ is denoted as the total target.*

We note that $q$ is not the probability in any case, since it extends the original probability $p$ over $\mathcal{O}$ by further considering the aggregations over $\mathcal{L}$. However, it is also sufficient to define objective functions to minimize the differences between total prediction and total target, which is SEAL regularization.

**Definition 2 (SEAL regularization)** *Given input $\boldsymbol{x}$, target $y$, model $f_\theta$, and a SEAL extension $(\mathcal{O}, \mathcal{L}, \mathbf{A}_1, \mathbf{A}_2, \alpha)$, the SEAL regularization is defined as $\phi(f_\theta(\boldsymbol{x}), \delta_y) = D(q(f_\theta(\boldsymbol{x})), q(\delta_y))$, where $D$ is a distance function.*

In this paper, we consider SEAL regularization where $D$ is the weighted $\ell_1$ metric:

$$\phi(f_\theta(\boldsymbol{x}), \delta_y) = D(q(f_\theta(\boldsymbol{x})), q(\delta_y)) = \sum_{s \in \mathcal{O} \cup \mathcal{L}} w_s |[q(f_\theta(\boldsymbol{x}))]_s - [q(\delta_y)]_s|, \tag{4}$$

where $w_s$ is the weight for each observed or latent label.

We have presented the basic framework of SEAL, then we detail how SEAL is used to explore label hierarchy and improve learning in the next parts.

### 3.3 LABEL HIERARCHY EXPLORATION WITH SEAL

In this section, we explore the label hierarchy under the SEAL extension $(\mathcal{O}, \mathcal{L}, \mathbf{A}_1, \mathbf{A}_2, \alpha)$. To achieve this, we first specify $\mathcal{L}$, $\mathbf{A}_1$, $\mathbf{A}_2$, and $\alpha$, which breakdowns into two tasks. The first task is to specify $\mathcal{L}$ and $\mathbf{A}_1$ to define the prior structure inside $\mathcal{L}$, while the second task is to specify how $\alpha$ and $\mathbf{A}_2$ are related to defining how the structure is optimized.

**Task (a): Prior Structure Specification.** To specify the prior structure inside $\mathcal{L}$, we choose $\mathbf{A}_1$ to be a trivial binary tree or trees derived from prior knowledge such as a part of a knowledge graph or a decision tree. This choice of $\mathbf{A}_1$ allows us to control the prior structure of the label hierarchy and incorporate prior domain knowledge. Additionally, we can use the hierarchical structure of $\mathbf{A}_1$ to guide the training of the model to improve performance.

**Task (b): Structure Optimization Specification.** To specify how $\alpha$ and $\mathbf{A}_2$ are related to defining how the structure is optimized, we note that $\alpha$ and $\mathbf{A}_2$ both reflect how the structure interacts with the model and data. Specifically, we compute $\alpha$ from $\mathbf{A}_2$ from the Markov chain on trees. This choice of $\alpha$ emphasizes more on the prediction of the model while $\mathbf{A}_2$ emphasizes more on the interpretation of the label hierarchy.

In summary, by specifying $\mathcal{L}$, $\mathbf{A}_1$, $\mathbf{A}_2$, and $\alpha$, we can explore the label hierarchy under the SEAL extension $(\mathcal{O}, \mathcal{L}, \mathbf{A}_1, \mathbf{A}_2, \alpha)$. This approach allows us to incorporate prior domain knowledge and guide the training of the model to improve performance while also providing a framework for interpreting the label hierarchy.

**Random Walk Construction of $\alpha$.** We observe that the matrix $\mathbf{A}$ satisfies the conditions in Theorem 2, and can be viewed as a Markov transition matrix on $\mathcal{L} \cup \mathcal{O}$ that follows the top-down direction over a tree. Therefore, the probability of a random walk from a node $s \in \mathcal{L}$ to a node $r \in \mathcal{O}$ can be computed by simulating the Markov chain. We define $\alpha_{sr}$ to be the probability of random walks starting from $s$ and ending at $r$, which can be interpreted as the probability that node $r$ is contained in the subtree of node $s$. Specifically, we have:

$$\alpha_{sr} = \left[ \sum_{k=1}^{\infty} \mathbf{A}^k \right]_{sr} = \left[ (\mathbf{I} - \mathbf{A})^{-1} \right]_{sr}, \tag{5}$$

where $\mathbf{I}$ is the identity matrix.

Moreover, we can further simplify the above equation by noting that $(\mathbf{I} - \mathbf{A})^{-1}$ can be precomputed. Specifically, we have: $(\mathbf{I} - \mathbf{A})^{-1} = \begin{pmatrix} (\mathbf{I} - \mathbf{A}_1)^{-1} & (\mathbf{I} - \mathbf{A}_1)^{-1} \mathbf{A}_2 \\ 0 & \mathbf{I} \end{pmatrix}$.

This construction of $\alpha$ is based on the Markov chain on trees and has been applied in other applications such as document distance (Takezawa et al., 2021) and hierarchical node clustering (Zügner et al., 2021). However, it is the first time that this construction has been used to train a deep neural network.

**Optimizing SEAL regularization** Once $\alpha$ is defined explicitly through $\mathbf{A}_2$, the expression of SEAL regularization $\Phi$ is also well defined. It simplifies to

$$\phi(f_\theta(\boldsymbol{x}), \delta_y) = (w^\top \left( (\mathbf{I} - \mathbf{A}_1)^{-1} \mathbf{A}_2 \mathbf{I} \right) (f_\theta(\boldsymbol{x}) - \delta_y))^{|\cdot|}, \tag{6}$$

where $|\cdot|$ denotes taking the element-wise absolute value. We set $w = \mathbf{1}_N$ for simplicity.

One could jointly optimize $\theta$ and $\mathbf{A}_2$ (or $\alpha$). Particular attention should be paid to $\mathbf{A}_2$ since it is discrete and required to satisfy the conditions in Theorem 2, making the optimization very hard. In this paper, we relax $\mathbf{A}_2 \in \{0,1\}^{(N-K) \times K}$ to $\mathbf{A}_2^{\text{soft}} \in [0,1]^{(N-K) \times K}$. One could employ projected gradient descent on each column of $\mathbf{A}_2^{\text{soft}}$ to ensure those conditions. More investigation on optimization can be found in Appendix A.6.

For clarity, we denote the SEAL regularization as $\phi_\Theta(f_\theta(\boldsymbol{x}), \delta_y)$, where the suffix $\Theta$ denotes the parameters defining latent hierarchy.

**Interpreting SEAL results** After $\mathbf{A}_2^{\text{soft}}$ is optimized, we can reconstruct $\mathbf{A}_2$ to interpret the explored label hierarchy. Specifically

$$(\mathbf{A}_2)_{sr} = \begin{cases} 1 & \text{if } s = \arg\max_{k \in \mathcal{L}} (\mathbf{A}_2^{soft})_{kr} \\ 0 & \text{otherwise} \end{cases} \tag{7}$$

Then the matrix $\mathbf{A}$ is derived after optimization.

### 3.4 LEARNING WITH SEAL

We have already defined SEAL regularization based on the model output $f_\theta(\boldsymbol{x})$ and the target $y$. Then it is natural to apply SEAL regularization to various learning scenarios.

We consider the learning process in a typical mini-batch setting. Given a batch of samples $B \subset D$, we consider the averaged summation $\Phi$ of SEAL regularization $\phi$ over the batch.

$$\Phi(\theta, \Theta; B) = \frac{1}{|B|} \sum_{(\boldsymbol{x}, y) \in B} \phi_\Theta(f_\theta(\boldsymbol{x}), \delta_y). \tag{8}$$

We note that $\delta_y$ could be the one-hot labels of the labeled data or pseudo-labels on unlabeled data.

#### 3.4.1 SEMI-SUPERVISED LEARNING WITH SEAL

Consider a general scheme (Gong et al., 2021) that unifies many prior semi-supervised algorithms. For the $n + 1$-th iteration, the model parameter is derived based on supervised loss $L(\theta)$ and consistency regularization $\Psi(\theta; \Theta)$. Specifically

$$\theta_{n+1} \leftarrow \arg\min_\theta \left\{ L(\theta) + \gamma \Psi(\theta; \theta_n) \right\}, \tag{9}$$

where $\theta_n$ denotes the model parameters at the $n$-th iteration and $\gamma$ is the loss balancing coefficient.

Adding SEAL to semi-supervised learning is no more than applying SEAL regularization to a supervised loss $L(\theta)$, which is shown in equation (16), and consistency regularization $\Psi(\theta, \Theta)$, which will be described below.

Usually speaking, the computation $\Psi(\theta; \theta_n)$ is conducted over a batch of unlabeled data $B^u \subset D^u$. For each sample $\boldsymbol{x} \in B^u$, the computation follows the following process:

1. **Pseudo-label prediction on weak augmentation** Computing the prediction of weakly-augmented image $\boldsymbol{x}' \in P_{\text{wAug}}(\boldsymbol{x})$ with model $f_{\theta_n}$ in the last iteration, which will be used to generate pseudo-labels in the next steps.
2. **Strong augmentation** For the input data $\boldsymbol{x}$, we sample strong argumentation $\boldsymbol{x}'' \in P_{\text{sAug}}(x)$.
3. **Selection** Some selection processes are applied to select the samples and assign them meaningful pseudo-labels $y'$s. This results in a new pseudo-labeled dataset $\hat{B}^u := \{(\boldsymbol{x}'', y')\}$.

Therefore, consistency regularization minimizes the differences between the prediction by the model and the pseudo-label $y'$, for example, using the cross entropy as follows:

$$\Psi(\theta, \theta_n) = \frac{1}{|\hat{B}^u|} \sum_{(\boldsymbol{x}'', y') \in \hat{B}^u} \text{CE}(f_{\theta_n}(\boldsymbol{x}''), \delta_{y'}). \tag{10}$$

Similar to equation (16), adding SEAL regularization is simply adding another term $\Phi(\theta, \Theta, \hat{B}^u)$. Then, we obtain the updating rule of semi-supervised learning with SEAL:

$$\theta_{n+1} \leftarrow \arg\min_\theta \left\{ L(\theta) + \gamma \left[ \Psi(\theta; \theta_n) + \lambda \Phi(\theta, \Theta; \hat{B}^u) \right] \right\}. \tag{11}$$

Many existing approaches (Sohn et al., 2020; Zhang et al., 2021; Wang et al., 2022) fit into this paradigm. So SEAL can be applied easily to such approaches.

**Summary for SEAL** We have demonstrated how to plugin SEAL with supervised and semi-supervised learning in equations (16) and (11), respectively. The key observation is that SEAL regularization can be applied as long as there is an objective function between model prediction $f(\boldsymbol{x})$ and target $y$.

## 4 THEORETICAL ANALYSIS OF SEAL REGULARIZATION

We find equation (6) has a similar structure to that of Tree-Wasserstein distance, so we shall first extend the definition of Tree-Wasserstein distance.

We shall first rewrite the Tree-Wasserstein distance's summation using the node as indices. Note the lower endpoint of each edge has a one-to-one correspondence with each node, thus we can see the weight of each edge as the weight of each node. Denote the tree as $\mathcal{X}$ and leaf nodes as $\mathcal{X}_{leaf}$. We can rewrite the expression of Tree-Wasserstein distance into $W_{d_{\mathcal{X}}}(\mu, \nu) = \sum_{v \in \mathcal{X}} w_v |\mu(\Gamma(v)) - \nu(\Gamma(v))|$. When $\mu$ and $\nu$ are supported only on the leaf set, we can rewrite $\Gamma(v)$ using the ancestor-child relationship. That is,

$$W_{d_{\mathcal{X}}}(\mu, \nu) = \sum_{v \in \mathcal{X}} w_v | \sum_{x \in \mathcal{X}_{\text{leaf}}} (\mu(x) - \nu(x)) \mathbb{I}_{\text{v is ancestor of x}} |. \tag{12}$$

If a directed graph has its adjacency matrix $A$ satisfying the conditions in Theorem 2 except relaxing the hard constraint $\{0, 1\}$ to $[0, 1]$, we shall call it a soft tree. Recall the subtree probabilistic interpretation of $\alpha$ in 3.3, we can define relaxed Tree-Wasserstein distance (RTW) as below.

**Definition 3 (Relaxed Tree-Wasserstein distance)** *Assume $\mathcal{X}$ is a soft tree and denote the leaf nodes as $\mathcal{X}_{leaf}$. For any two probability measures supported on $\mathcal{X}_{leaf}$. The relaxed tree Wasserstein distance $W_{d_{\mathcal{X}}}^{relax}(\mu, \nu)$ is given as follows:*

$$W_{d_{\mathcal{X}}}^{relax}(\mu, \nu) = \sum_{v \in \mathcal{X}} w_v \left| \sum_{x \in \mathcal{X}_{leaf}} \alpha_{vx} (\mu(x) - \nu(x)) \right|. \tag{13}$$

If we let $\mathcal{X}_{\text{leaf}}$ be the set of *observed* labels $\mathcal{O}$ and $\mathcal{X}$ be *total* labels $\mathcal{O} \cup \mathcal{L}$. We can then show the connection between the relaxed tree Wasserstein distance and the weighted total classification error given by equation (6).

**Theorem 3** *The weighted total classification loss described by Eqn. (6) under $\mathbf{A}_2^{soft}$ coincides with $W_{d_{\mathcal{X}}}^{relax}(f_\theta(\boldsymbol{x}), \delta_y)$.*

**Proof 1** *Please see Appendix B.2.*

## 5 APPLYING SEAL TO SEMI-SUPERVISED LEARNING

Firstly, SEAL improves standard supervised learning , details could be found in Appendix C. Then we present our major experimental results on semi-supervised learning. For details of dataset and implementations of semi-supervised learning, please refer to Appendix C.3.

### 5.1 BASELINES

The baselines we consider in our experiments are those prior works similar to FixMatch, such as $\Pi$-Model (Laine & Aila, 2016), Pseudo Label (Lee et al., 2013), Mean Teacher (Tarvainen & Valpola, 2017), MixMatch (Berthelot et al., 2019b), ReMixMatch (Berthelot et al., 2019a), VAT (Miyato et al., 2018), UDA (Xie et al., 2020), FlexMatch (Zhang et al., 2021)and DebiasPL (Wang et al., 2022). However, we find that our proposed method SEAL is lightweight yet effective and outperforms all of these baselines on all three datasets in nearly all settings.

### 5.2 FINDINGS

**SEAL is lightweight and effective.** SEAL is easy to implement and with the aid of SEAL and its variants, we can achieve state-of-art results on all three datasets under all label amount settings. Results can be found in Tables 1 and 2.

**The fewer labeled data, the more significant improvements.** Interestingly, we observe that the fewer labeled data available, the more significant gains we can achieve using SEAL. For instance, on CIFAR10, we obtain a remarkable 7.52 accuracy gain with only 40 labeled data, while we only see a 0.64 accuracy gain with 250 labeled data. This finding highlights the effectiveness of our proposed method in situations where labeled data is scarce.

Table 1: Error rates (100% - accuracy) on CIFAR-10. **bold** means the best.

| Dataset | CIFAR-10 | |
|---|---|---|
| # Label | 40 | 250 |
| Π Model (Rasmus et al., 2015a) | $74.34_{\pm1.76}$ | $46.24_{\pm1.29}$ |
| Pseudo Label (Lee et al., 2013) | $74.61_{\pm0.26}$ | $46.49_{\pm2.20}$ |
| VAT (Miyato et al., 2018) | $74.66_{\pm2.12}$ | $41.03_{\pm1.79}$ |
| MeanTeacher (Tarvainen & Valpola, 2017) | $70.09_{\pm1.60}$ | $37.46_{\pm3.30}$ |
| MixMatch (Berthelot et al., 2019b) | $36.19_{\pm6.48}$ | $13.63_{\pm0.59}$ |
| ReMixMatch (Berthelot et al., 2019a) | $9.88_{\pm1.03}$ | $6.30_{\pm0.05}$ |
| UDA (Xie et al., 2020) | $10.62_{\pm3.75}$ | $5.16_{\pm0.06}$ |
| FixMatch (Sohn et al., 2020) | $7.47_{\pm0.28}$ | $5.07_{\pm0.05}$ |
| Dash (Xu et al., 2021) | $8.93_{\pm3.11}$ | $5.16_{\pm0.23}$ |
| MPL (Pham et al., 2021) | $6.93_{\pm0.17}$ | $5.76_{\pm0.24}$ |
| FlexMatch (Zhang et al., 2021) | $4.97_{\pm0.06}$ | $4.98_{\pm0.09}$ |
| DebiasPL (Wang et al., 2022) | $5.40_{\pm1.30}$ | $5.60_{\pm0.10}$ |
| **SEAL** | $6.29_{\pm0.58}$ | $4.43_{\pm0.55}$ |
| **SEAL (Debiased)** | $4.66_{\pm0.07}$ | $4.41_{\pm0.18}$ |

Table 2: Error rates (100% - accuracy) on CIFAR-10, CIFAR-100, and STL-10 dataset of state-of-the-art methods for semi-supervised learning. **bold** means the best.

| Dataset | CIFAR-100 | | STL-10 | |
|---|---|---|---|---|
| # Label | 400 | 2500 | 40 | 250 |
| Π Model (Rasmus et al., 2015a) | $86.96_{\pm0.80}$ | $58.80_{\pm0.66}$ | $74.31_{\pm0.85}$ | $55.13_{\pm1.50}$ |
| Pseudo Label (Lee et al., 2013) | $87.45_{\pm0.85}$ | $57.74_{\pm0.28}$ | $74.68_{\pm0.99}$ | $55.45_{\pm2.43}$ |
| VAT (Miyato et al., 2018) | $85.20_{\pm1.40}$ | $46.84_{\pm0.79}$ | $74.74_{\pm0.38}$ | $56.42_{\pm1.97}$ |
| MeanTeacher (Tarvainen & Valpola, 2017) | $81.11_{\pm1.44}$ | $45.17_{\pm1.06}$ | $71.72_{\pm1.45}$ | $56.49_{\pm2.75}$ |
| MixMatch (Berthelot et al., 2019b) | $67.59_{\pm0.66}$ | $39.76_{\pm0.48}$ | $54.93_{\pm0.96}$ | $34.52_{\pm0.32}$ |
| UDA (Xie et al., 2020) | $46.39_{\pm1.59}$ | $27.73_{\pm0.21}$ | $37.42_{\pm8.44}$ | $9.72_{\pm1.15}$ |
| Dash (Xu et al., 2021) | $44.82_{\pm0.96}$ | $27.15_{\pm0.22}$ | $34.52_{\pm4.30}$ | - |
| MPL (Pham et al., 2021) | $46.26_{\pm1.84}$ | $27.71_{\pm0.19}$ | $35.76_{\pm4.83}$ | $9.90_{\pm0.96}$ |
| FixMatch (Sohn et al., 2020) | $46.42_{\pm0.82}$ | $28.03_{\pm0.16}$ | $35.97_{\pm4.14}$ | $9.81_{\pm1.04}$ |
| FlexMatch (Zhang et al., 2021) | $39.94_{\pm1.62}$ | $26.49_{\pm0.20}$ | $29.15_{\pm4.16}$ | $8.23_{\pm0.39}$ |
| **SEAL** | $45.18_{\pm0.79}$ | $26.99_{\pm0.23}$ | $33.27_{\pm3.21}$ | $9.77_{\pm0.54}$ |
| **SEAL (Cirriculum)** | $\mathbf{39.73_{\pm1.58}}$ | $\mathbf{26.39_{\pm0.29}}$ | $\mathbf{16.15_{\pm2.63}}$ | $\mathbf{7.92_{\pm0.47}}$ |

**SEAL can be boosted by various techniques.** Moreover, we demonstrate that our proposed method can be further enhanced by incorporating various existing semi-supervised learning techniques, such as **Curriculum** Pseudo Label (Zhang et al., 2021) and **Debiased** Pseudo Label (Wang et al., 2022), into SEAL framework with minimal effort. This implies that any future work on improving the quality of pseudo labels can be easily adapted into our SEAL framework.

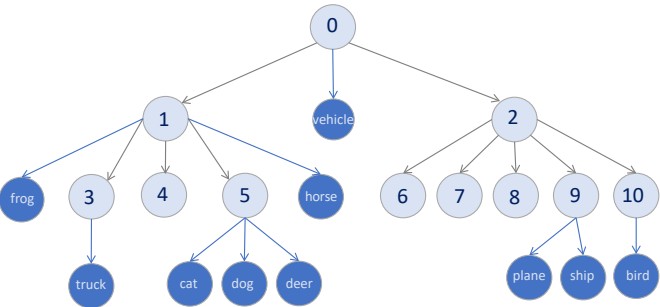

Figure 2: A hierarchy of posterior tree structure

**Meaningful posterior tree structure.** We present a learned tree structure in Figure 2. We can find semantically close classes are indeed close on the tree. For example, animals cat, dog, and deer are all child nodes of node 5.

# 6 RELATED WORK

## 6.1 SEMI-SUPERVISED LEARNING

Most existing methods for consistency regularization aim to improve the quality of the pseudo labels generated from unlabeled data. For example, SimPLE (Hu et al., 2021) introduces a paired loss that minimizes the statistical distance between confident and similar pseudo labels. Dash (Xu et al., 2021) and FlexMatch (Zhang et al., 2021) propose dynamic and adaptive strategies for filtering out unreliable pseudo labels during training. MaxMatch (Li et al., 2022) proposes a worst-case consistency regularization technique that minimizes the maximum inconsistency between an original unlabeled sample and its multiple augmentations with theoretical guarantees. A notable exception is SemCo (Nassar et al., 2021), which leverages external label semantics to prevent the deterioration of pseudo label quality for visually similar classes in a co-training framework.

Though proposed in different techniques, all these methods rely on a fixed objective function to define the consistency, which is usually the cross-entropy over the label space. Our work differs from these methods by proposing a novel way to extend the cross-entropy with latent labels and hierarchical structures. Therefore, our method can complement existing methods whenever cross-entropy is used.

## 6.2 LABEL HIERARCHIES

Label relationships are essential prior knowledge for improving model performance, and can be represented by semantic structures among labels. One prominent form of label structure is the label hierarchy (Garnot & Landrieu, 2020), which can be obtained from external sources like decision trees (Wan et al., 2020) and knowledge graphs (Miller, 1998; Speer et al., 2017). This information can be leveraged to train models as semantic embeddings (Deng et al., 2010; 2012; Barz & Denzler, 2019; Liu et al., 2020; Wang et al., 2021b; Karthik et al., 2021; Nassar et al., 2021) or objective functions (Bertinetto et al., 2020; Bilal et al., 2017; Garnot & Landrieu, 2020; Garg et al., 2022b). Additionally, the hierarchical information can also be incorporated as part of the model structure (Kontschieder et al., 2015; Wu et al., 2016; Wan et al., 2020; Chang et al., 2021; Garg et al., 2022a).

Pre-defined label hierarchies are widely acknowledged as an essential source of prior knowledge for the label space in classification. This has been extensively discussed in the literature. The label hierarchy information can be used to improve the model training process, such as by embedding the hierarchical labels to maximize the similarity between the latent embedding of the input image and the embedding of its label (Bengio et al., 2010; Deng et al., 2012; Frome et al., 2013). This idea has been generalized to various embedding spaces (Barz & Denzler, 2019; Liu et al., 2020; Garnot & Landrieu, 2020) and the joint learning scheme where image embeddings and label embeddings are both optimized (Wu et al., 2016; Chang et al., 2021). Additionally, hierarchical structures can also be explicitly used to make the training process hierarchical-aware (Deng et al., 2010; Bilal et al., 2017; Bertinetto et al., 2020; Karthik et al., 2021; Garg et al., 2022a;b). However, existing work typically treats the hierarchical label structure as prior knowledge. In contrast, our approach leverages the posterior latent label structures given the presence of labeled and unlabeled samples.

# 7 CONCLUSION AND FUTURE WORK

In this paper, we propose a framework SEAL to jointly train the model of high performances and the label structure of significance. The SEAL framework is flexible to be adapted to various learning schemes, and can even incorporate the prior structure given by the external knowledge and the information given by the data. Experimental results support the effectiveness of the SEAL framework. Theoretical understanding of SEAL via optimal transport theory is also discussed. Future works may include incorporating more complex prior knowledge or applying the SEAL framework to self-supervised learning.

## REPRODUCIBILITY STATEMENT

To foster reproducibility, we submit our experiment code as supplementary material.

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

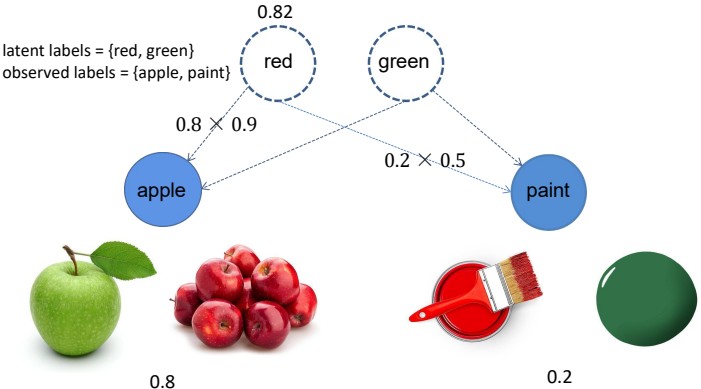

Figure 3: An illustrative example

# A APPENDIX

## A.1 A PICTORIAL VIEW OF EXAMPLE 1

We present the Figure used in the motivation section.

## A.2 ABLATION STUDIES

In this section, we focus on analyzing the influence of different parameters on the performance of our proposed method, SEAL (Debiased), using the CIFAR10 dataset with only 40 labeled samples.

### A.2.1 DIFFERENT TREE STRUCTURE

| Tree Name | Acc. on CIFAR10 (40 labels) |
|---|---|
| Without Tree | 94.60 |
| Trivial Tree | 95.18 |
| Random Tree | 95.34 |
| NBDT (Wan et al., 2020) Tree | 95.39 |

Table 3: Classification under different tree structure

Next, we examine how the choice of tree structure affects the results. We compare three different trees: a trivial tree with all 10 classes as leaf nodes besides one root node, a randomly generated depth-4 tree with 21 nodes, and the NBDT tree proposed in (Wan et al., 2020), which has a well-designed hierarchy as shown in Figure 7. We use the adjacency matrix of the internal nodes induced subtree as the adjacency matrix $A_1$ in our method for each tree.

The results of the classification accuracies under these different trees are presented in Table 3. As we can see from the table, using tree structures consistently improves the classification accuracies compared to the vanilla cases. It is worth noting that the NBDT tree, which is carefully designed, achieves the highest accuracy, while our randomly generated tree performs better than the other cases.

### A.2.2 DIFFERENT REGULARIZER

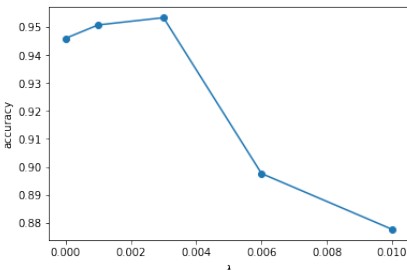

Figure 4: The influence of different hyper-parameter

Lastly, we explore the impact of the regularizer on classification accuracy by experimenting with 5 different $\lambda$ values, including the case of $\lambda = 0$ to illustrate the necessity of the regularizer. As shown in Figure 4, when $\lambda$ approaches 0.003, the accuracy increases, while it decreases as $\lambda$ deviates from 0.003. Therefore, we conclude that the optimal value of $\lambda$ is around 0.003. We note that the accuracy drops significantly when $\lambda$ is too large, which may be attributed to the imbalance in loss scale.

### A.3 DIFFERENT THRESHOLD

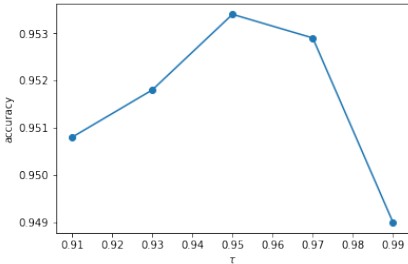

Figure 5: Varying different threshold for pseudo labeling

We first investigate the impact of threshold choice on the accuracy of our method. We experiment with five different values of $\tau$ and plot the results in Figure 5. As shown in the figure, the accuracy increases as $\tau$ approach 0.95, and decreases when $\tau$ deviates from 0.95. This suggests that choosing a threshold of 0.95 yields the best performance.

### A.4 CONVERGENCE SPEEDUP

Figure 6 displays the Top-1 accuracy of CIFAR100-2500 labels, showcasing that with the aid of SEAL, convergence becomes faster and more stable. With the addition of SEAL, the model consistently outperforms the initial training process in all epochs.

### A.5 COMPUTATION OVERHEAD OF INTRODUCING SEAL

We investigate the additional computation time required when applying SEAL to our method. Table 4 shows the results of these experiments, which were conducted using an Nvidia GeForce RTX 2080 Ti. We observe that the computation of SEAL incurs only a marginal increase in the computation time, demonstrating its efficiency in practice.

### A.6 DIFFERENT UPDATING RULE FOR THE ADJACENCY MATRIX

Efficient updating of the soft adjacency matrix $\mathbf{A}_2$ is essential in the experiments. Two popular approaches have been used for updating $\mathbf{A}_2$. One approach is to consider each column of $\mathbf{A}_2$

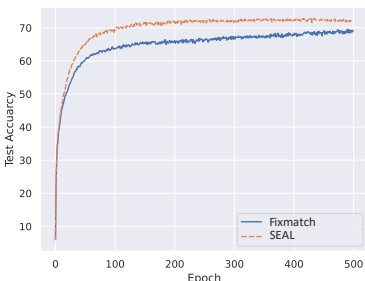

Figure 6: Top-1 accuracy on CIFAR100 dataset (2500 labels)

Table 4: Computation Time of 1 epoch

| Update Method | Computation Time (Minutes) |
| --- | --- |
| with SEAL | 7.11 |
| without SEAL | 7.10 |

as the realization of softmax mapping. The other approach is to use projected gradient descent (PGD) to update the matrix, projecting each column onto the probability simplex. The classification accuracies of both methods are summarized in Table 5.

Table 5: Different methods for updating adjacency matrix

| Update Method | CIFAR10-40 labels |
| --- | --- |
| PGD | 95.34 |
| Softmax | 86.46 |

It is evident that PGD has a significant advantage over the softmax mapping-based approach. This observation is also reported in (Zügner et al., 2021). The poor performance of the softmax-based approach may be attributed to a bad initialization, where the optimization is trapped by the bad starting point.

Fig. 7 shows the hierarchy of the NBDT tree.

## B  More Theoretical Results

### B.1  Optimal Transport

**Definition 4 (Wasserstein-1 distance)** *Consider two probability distribution: $\boldsymbol{x} \sim \mu$, and $\boldsymbol{y} \sim \nu$. The Wasserstein-1 distance between $\mu$ and $\nu$ can be defined as:*

$$\mathcal{W}(\mu, \nu) = \min_{\pi \in \Pi(\mu, \nu)} \int_{\mathcal{X} \times \mathcal{X}} c(\boldsymbol{x}, \boldsymbol{y}) \, d\pi,$$

*where $\mathcal{X}$ is the space that $\mu$ and $\nu$ supported on, $c(\cdot, \cdot)$ is a cost function defined on the cartesian space $\mathcal{X} \times \mathcal{X}$, and $\Pi(\mu, \nu)$ is the set of all possible couplings of $\mu$ and $\nu$ ; and $\pi$ is a joint distribution satisfying $\int_{\mathcal{X}} \pi(\boldsymbol{x}, \boldsymbol{y}) \, d\boldsymbol{y} = \mu(\boldsymbol{x})$ and $\int_{\mathcal{X}} \pi(\boldsymbol{x}, \boldsymbol{y}) \, d\boldsymbol{x} = \nu(\boldsymbol{y})$.*

### B.2  The Relation between SEAL and RTW

Note that we assume no correlation between real labels, thus the vector constructed by $\Pr(r|y)$ as its $r$-th component is exactly $\delta_y$.

As the nodes in $\mathcal{O} \cup \mathcal{L}$ are exactly all the *total* labels, and those in $\mathcal{O}$ are all the *observed* labels. Comparing equations (6) and (13) yields the result. For convenience, we consider the entire tree

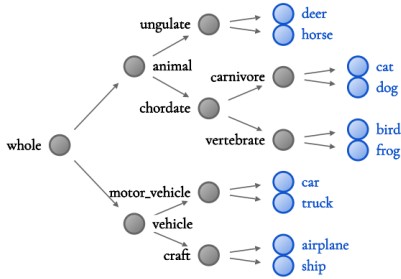

Figure 7: The hierarchy of NBDT Tree

metric space as $\mathcal{X}$. Then it is easy to see that $\mathcal{X} = \mathcal{O} \cup \mathcal{L}$, and $\mathcal{X}_{\text{leaf}} = \mathcal{O}$ are the leaf nodes of the tree.

### B.3 BASIC PROPERTY OF RTW

The positive definiteness and symmetry of $W_{d_{\mathcal{X}}}^{\text{relax}}$ is clear from it's definition. Then we show it satisfies the triangular inequality. For any probability measures $\lambda$, $\mu$ and $\nu$ on $\mathcal{X}_{\text{leaf}}$.

$$
\begin{aligned}
& W_{d_{\mathcal{X}}}^{\text{relax}} \left(\lambda, \mu\right) + W_{d_{\mathcal{X}}}^{\text{relax}} \left(\mu, \nu\right) \\
&= \sum_{v \in \mathcal{X}} w_v \left| \sum_{x \in \mathcal{X}_{\text{leaf}}} \alpha_{vx} \left(\lambda(x) - \mu(x)\right) \right| \\
&\quad + \sum_{v \in \mathcal{X}} w_v \left| \sum_{x \in \mathcal{X}_{\text{leaf}}} \alpha_{vx} \left(\mu(x) - \nu(x)\right) \right| \\
&\geq \sum_{v \in \mathcal{X}} w_v \left| \sum_{x \in \mathcal{X}_{\text{leaf}}} \alpha_{vx} \left(\lambda(x) - \nu(x)\right) \right| \\
&= W_{d_{\mathcal{X}}}^{\text{relax}} \left(\lambda, \nu\right).
\end{aligned}
\tag{14}
$$

Note that when $\mathcal{X}$ is a hard tree, we have $\mathbb{I}_{\text{v is the ancestor of x}} = \alpha_{vx}$. Then from equations (12) and (13), we know that RTW is exactly the tree Wasserstein distance in this degenerate case.

Here we would present some theoretical properties of relaxed Tree-Wasserstein distance next to illustrate why it is a good metric defined on trees.

**Theorem 4** $W_{d_{\mathcal{X}}}^{relax} \left(\cdot, \cdot\right)$ *defines a metric on the probability space. Furthermore, when* $\mathbf{A}$ *is the (hard) adjacency matrix of a tree, the relaxed tree Wasserstein distance is exactly the tree Wasserstein distance.*

**Proof 2** *Please see Appendix B.3.*

**Theorem 5** *The relaxed tree Wasserstein distance is a negative definite kernel.*

**Proof 3** *Please see Appendix B.4.*

### B.4 KERNEL PROPERTY OF RTW

**Definition 5** *(Berg et al., 1984) A function* $k : \mathcal{M} \times \mathcal{M} \to \mathbb{R}$ *is negative definite if for* $\forall n \geq 2$, $\forall x_1, x_2, \ldots, x_n \in \mathcal{M}$ *and* $\forall c_i \in \mathbb{R}$ *such that* $\sum_{i=1}^{n} c_i = 0$, *we have* $\sum_{i,j} c_i c_j k \left(x_i, x_j\right) \leq 0$.

We shall prove that RTW is a negative definite kernel on the tree leaf Wasserstein space $\mathcal{M} = \mathbb{P}(\mathcal{X}_{\text{leaf}})$. We define a mapping $\Phi$ where

$$
\Phi(x) = \left( \begin{matrix} \left(\mathbf{I} - \mathbf{A}_1\right)^{-1} \mathbf{A}_2 \\ \mathbf{I} \end{matrix} \right) x.
\tag{15}
$$

Table 6: ViN best knn classification

| initial feature | smooth feature | SEAL feature | prob space hard treedis |
|---|---|---|---|
| 0.6725 | 0.6818 | 0.6831 | 0.8094 |

Table 7: ResNet18 best knn classification

| initial feature | smooth feature | SEAL feature | prob space hard treedis |
|---|---|---|---|
| 0.9542 | 0.9551 | 0.9572 | 0.9574 |

Note $k(x_i, x_j) = \|w \circ \Phi(x_i) - w \circ \Phi(x_j)\|_1$. Since the definition of negative definiteness is only related to the value of $k$, we can transform the problem of considering $\Phi$ only. Note $k$ is only a weighted $l_1$ distance between $\Phi$, from the separability of $l_1$ norm and (Le et al., 2019)'s Lemma A.2, it is clear that RTW is negative definite.

## C  A CLOSER LOOK AT THE SUPERVISED SETTINGS

In this section, we shall show the performance of SEAL regularization on two backbones. One is a backbone with fewer parameters, another is the standard ResNet 18 backbone.

Consider a supervised learning objective $L(\theta)$ over a batch, such as Cross-Entropy (CE) to train the neural network $\theta$. One could derive the SEAL regularized objective as

$$\mathcal{L}(\theta, \Theta) = \frac{1}{|B|} \sum_{(\boldsymbol{x}, y) \in B} \mathrm{CE}(f_\theta(\boldsymbol{x}), \delta_y)) + \lambda \phi(f_\theta(\boldsymbol{x}), \delta_y))$$
$$= L(\theta) + \lambda \Phi(\theta, \Theta; B) \qquad (16)$$

Optimizing $\mathcal{L}(\theta, \Theta)$ jointly trains the neural network $f_\theta$ and the latent hierarchy defined by $\Theta$.

### C.1  VIN BACKBONE

We train Vision Nystromformer (ViN) (Jeevan & Sethi, 2021) with optimizer AdamW for 60 epochs and get classification accuracy 66.65%, apply the same configuration to label smoothing will give an accuracy of 68.02%. SEAL regularization boosts the accuracy of initial ViN from 66.65% to 68.52% within 5 epochs.

We are also interested in t-SNE visualization of the learned backbone feature, Fig. 9a is the initial ViN feature, and Fig. 9b ViN with label smoothing $\alpha = 0.1$, Fig. 9c is the learned ViN feature with SEAL regularization. Fig. 9d is slightly different, and we use the learned relaxed Tree-Wasserstein distance on probability space as the similarity metric.

As for the k-nearest neighbors (kNN) task, we choose the best $k$ for each subtask respectively. We summarize the result in Table 6. Note relaxed Tree-Wasserstein distance is a well-defined metric, so we also calculate the relaxed Tree-Wasserstein distance on probability space to do the knn task. The tree we used is plotted in Fig. 8a. The tree shows some semantic relations between classes, as semantic closer classes have smaller tree distances.

### C.2  RESNET18 BACKBONE

Inspired by paper NBDT, on CIFAR10 we train 200 epochs, and the origin (trained by cross-entropy loss) accuracy is 95.42%. With label smoothing, the accuracy is 95.54%, while with SEAL regularization, the accuracy is 95.75%. In the above experiments, we train the first 180 epochs using the same loss as the initial and turn the loss to label smoothing or RTW respectively.

We are also interested in t-SNE visualization of the learned backbone feature, Fig. 10a is the initial VIN feature, and Fig. 10b VIN with label smoothing $\alpha = 0.1$, Fig. 10c is the SEAL boosted VIN

feature. Fig. 10d is slightly different, and we use the learned tree distance on probability space as the similarity criteria.

As for the k-nearest neighbors (kNN) task, we summarize the result in Table 7. Note relaxed Tree-Wasserstein distance is a well-defined metric, so we also calculate the relaxed Tree-Wasserstein distance on probability space to do the knn task. The tree we used is plotted in Figure 8b. The tree shows some semantic relations between classes, as semantic closer classes have smaller relaxed Tree-Wasserstein distances.

Note that in the original paper NBDT, the initial accuracy is 94.97%, their method gets 94.82%. Their initial accuracy is slightly lower than ours may be due to the number of epochs they run being smaller.

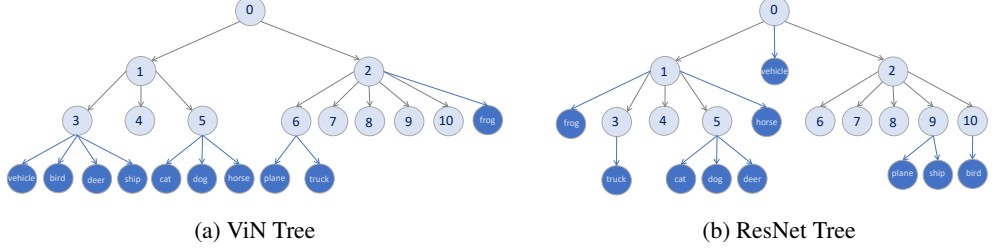

(a) ViN Tree                                             (b) ResNet Tree

Figure 8: Comparison of ViN Tree and ResNet Tree

### C.3 SEMI-SUPERVISED DETAILS

#### C.3.1 DATASETS

We evaluate our proposed method on three popular datasets, namely CIFAR10, CIFAR100, and STL-10.

**CIFAR10 and CIFAR100.** CIFAR10(Krizhevsky et al., 2009) contains 60,000 colored images in 10 different classes, where each image has a size of $32 \times 32$ pixels. The training set consists of 50,000 labeled images and the test set consists of 10,000 labeled images. Similarly, CIFAR100(Krizhevsky et al., 2009) contains 100 different classes with the same image size and a similar number of images.

**STL-10.** STL-10(Coates et al., 2011) is a semi-supervised benchmark that contains 10 classes. It is adapted from ImageNet(Deng et al., 2009) and contains 500 labeled training samples and 800 labeled testing samples per class. Additionally, it has 10,000 unlabeled images, some of which are not from the labeled classes.

Following the standard semi-supervised learning setting to sample, we sample the labeled images equally and randomly from all classes. To ensure statistical significance, we repeat each experiment five times and calculate the mean and standard deviation of the results.

#### C.3.2 IMPLEMENTATION OF SEAL

**Defining SEAL extension** $(\mathcal{O}, \mathcal{L}, \mathbf{A}_1, \mathbf{A}_2, \alpha)$. For fair comparison and injecting no prior knowledge, $\mathbf{A}_1$ and $\mathbf{A}_2^{soft}$ are both randomly initialized. For CIFAR10 and STL-10, $K = 10$ and $N = 21$. For CIFAR100, $K = 100$ and $N = 130$. More ablations and details can be found in Section A.2.1.

**Combining SEAL with other learning methods.** The default setting of SEAL for semi-supervised learning is adopted from the same configuration and hyper-parameters used in FixMatch(Sohn et al., 2020). SEAL (Curriculum) adopts the curriculum pseudo-labeling technique and hyper-parameters used in FlexMatch(Zhang et al., 2021). SEAL (Debiased) adopts the debiasing trick and hyper-parameters used in DebiasPL(Wang et al., 2022).

**Optimizing with SEAL regularization.** Specifically, we use a (batch) stochastic gradient descent (SGD) optimizer with a momentum of 0.9. We set the learning rate scheduler as the cosine decay scheduler, where the learning rate $\beta$ can be expressed as $\beta = \beta_0 \mathbf{cos}(\frac{7\pi}{16} \frac{s}{S})$. Here, $\beta_0$ is the initial learning rate set to 0.03, $s$ is the current optimization step, and $S$ is the total number of optimization

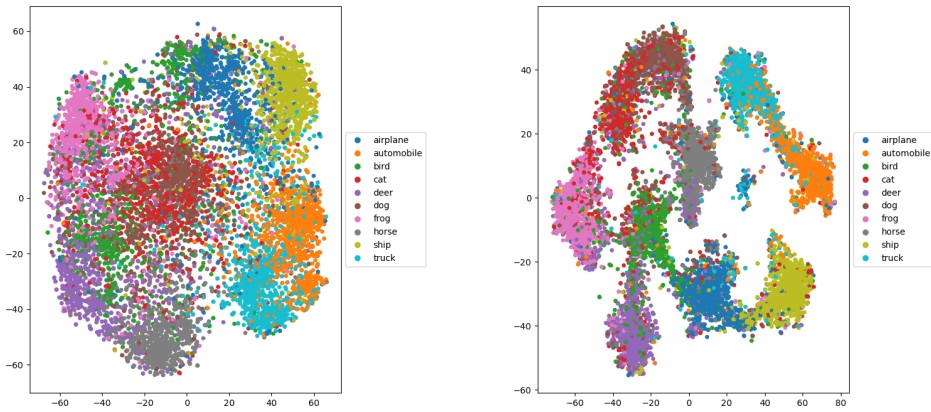

(a) t-SNE visualization of the initial feature of ViN backbone model.

(b) t-SNE visualization of the learned feature of ViN backbone model with label smoothing.

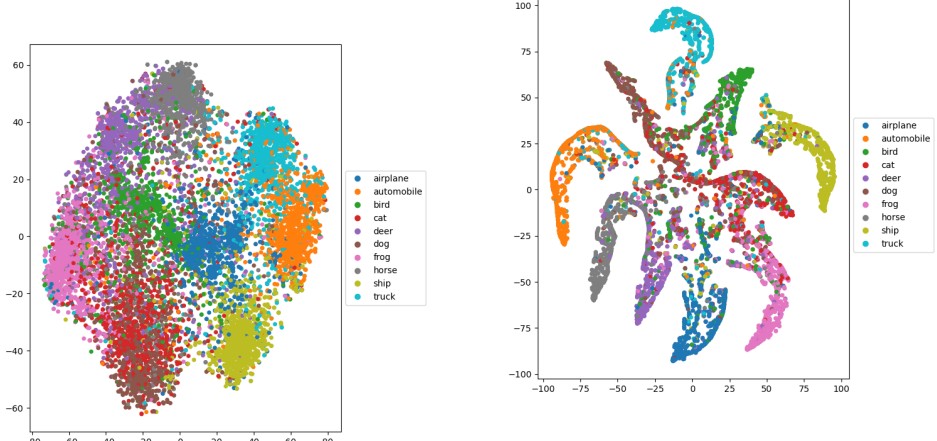

(c) t-SNE visualization of the learned feature of ViN backbone model with SEAL regularization.

(d) t-SNE visualization of the learned probability of ViN model with SEAL regularization. The similarity metric is defined by relaxed Tree-Wasserstein distance.

Figure 9: Comparison of t-SNE visualizations of features and probabilities learned by the ViN model with different regularization methods. (a) shows the initial feature of the ViN backbone model. (b) shows the learned feature with label smoothing. (c) shows the learned feature with SEAL regularization. (d) shows the learned probability with SEAL regularization using relaxed Tree-Wasserstein distance as the similarity metric.

steps set to $2^{20}$. We set the batch size of the labeled training data to 64, and the ratio of unlabeled training data to labeled data $\mu$ is set to 7. We set the threshold $\tau$ to 0.95, and the weak and strong augmentation functions used in our experiments are based on RandAugment(Cubuk et al., 2020). We use WideResNet-28-2 as the backbone model for our experiments.

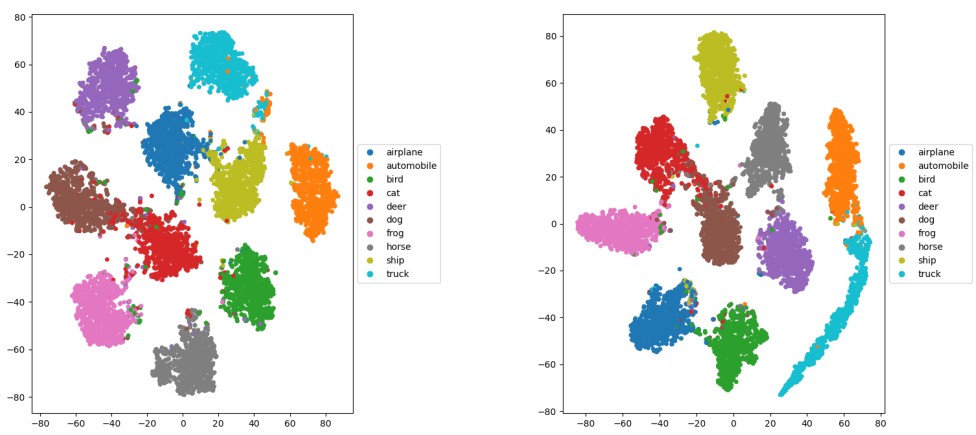

(a) t-SNE visualization of the initial feature of ResNet18 backbone model

(b) t-SNE visualization of the learned feature of ResNet18 backbone model with label smoothing

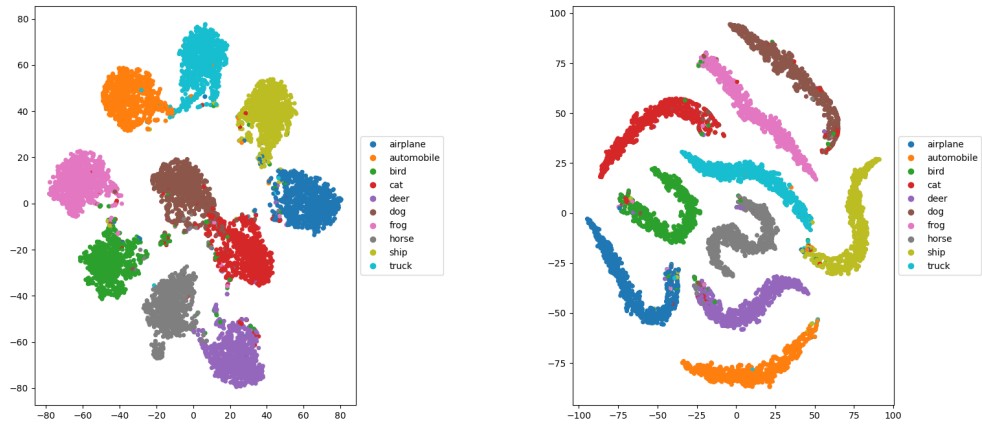

(c) t-SNE visualization of the learned feature of ResNet18 backbone model with SEAL regularization

(d) t-SNE visualization of the learned probability of ResNet18 backbone model with SEAL regularization. Similarity metric is defined by relaxed Tree-Wasserstein distance.

Figure 10: Visualizations of ResNet18 backbone model features with different regularization methods.

