# OpenReview forum: "SEAL: Simultaneous Label Hierarchy Exploration And Learning"
_ICLR.cc/2024/Conference — Submitted to ICLR 2024_

### Official Review · Reviewer_DPWp · 2023-10-25

**Soundness:** 3 good
**Presentation:** 2 fair
**Contribution:** 2 fair
**Rating:** 5
**Confidence:** 3

**Summary:**

The authors propose to exploit a data-driven label hierarchical structure and perform semi-supervised learning simultaneously. Specifically, they introduce latent labels that are aligned with observed labels to capture the data-driven label hierarchy. And a Wasserstein-based regularization term is designed to encourage agreement on both observed and latent label alphabets. Experiments are conducted in both supervised and semi-supervised scenarios.

**Strengths:**

Exploring and incorporating the label hierarchy structure for semi-supervised learning is an intriguing area of investigation. Furthermore, I think the design of ‘SEAL extension’ is very subtle and interesting.

**Weaknesses:**

1.  While the former sections of the paper are exceptionally well-written, the later parts, particularly the approach section, pose challenges in terms of comprehension；
2.  It is challenging to grasp the underlying meaning of 'latent labels' in the explored hierarchical structure. For instance, 'red' and 'green' appear to be related to both 'apple' and 'paint,' even though the author provided Example 1；
3.  Following the previous question, it is important to note that in the current method, which is based on a hierarchical, the calibration of predictions at each level becomes crucial. However, the authors currently seem to neglect the calibration at latent levels;
4.  The theoretical analysis presented lacks valuable insights for the proposed approach. Furthermore, the author fails to clearly and concisely describe the integration of theory with their proposed approach, as mentioned in the Weaknesses 1;
5.  The datasets used in the paper are very small and trivial, and the comparative methods employed are not up to date with the SOTA. The experiments are not convincing enough to demonstrate the effectiveness of the proposed approach.

**Questions:**

Please see the above.

---

> ### Author Response · Authors · 2023-11-22
> **Response to reviewer DPWp**
>
> >Q1: While the former sections of the paper are exceptionally well-written, the later parts, particularly the approach section, pose challenges in terms of comprehension；
>
> A1: Thank you for your suggestion. We will adjust our paper accordingly based on your suggestion.
>
> >Q2: It is challenging to grasp the underlying meaning of 'latent labels' in the explored hierarchical structure. For instance, 'red' and 'green' appear to be related to both 'apple' and 'paint,' even though the author provided Example 1；
>
> A2: Thank you for your feedback. The latent labels can be seen as "superclasses" or "higher order abstraction of classes". The latent labels can be seen as capturing the underlying relationships among classes. In our example, 'red' and 'green' indeed to be related to both 'apple' and 'paint,'.  **These two latent labels capture the relationship that apples (or paints) may have both colors green and red.**
>
> >Q3: Following the previous question, it is important to note that in the current method, which is based on a hierarchical, the calibration of predictions at each level becomes crucial. However, the authors currently seem to neglect the calibration at latent levels;
>
> A3: Thank you for your question. When you have a fixed hierarchy, you can calibrate the prediction at each level. But our adaptively updates the hierarchy to make the tree structure more suitable for the data distribution. **As the hierarchical level may change in our SEAL method, we define the total prediction error on the entire hierarchy to deal with this problem.**
>
> >Q4: The theoretical analysis presented lacks valuable insights for the proposed approach. Furthermore, the author fails to clearly and concisely describe the integration of theory with their proposed approach, as mentioned in the Weaknesses 1;
>
> A4: Thank you for your feedback. The theoretical analysis shows that our new loss can be interpreted as an optimal transport loss. As cross-entropy loss can be seen as a metric measures the similarity between probability distributions, **thus the theoretical analysis makes adding our new loss as a regularizer to cross-entropy loss in our SEAL method is more natural.**
>
>
> >Q5: The datasets used in the paper are very small and trivial, and the comparative methods employed are not up to date with the SOTA. The experiments are not convincing enough to demonstrate the effectiveness of the proposed approach.
>
> A5: Thank you for your question. Firstly, we want to point out that the benchmarks we used are widely adopted in SSL literatures, so conduct experiments on these benchmarks may give a fair comparision. **Secondly, our method is easy to implement, fast, easy to plug in existing SOTA to further increase accuracy. So we think this demonstrates its effectiveness.** Lastly, as the reviewer questions "the comparative methods employed are not up to date with the SOTA". We conduct additional experiments on a recently proposed SOTA FreeMatch and results are summarized in the following Table (in error rate).
>
> |  | CIFAR-10: 40 | CIFAR-10: 250 | CIFAR-100: 400 | CIFAR-100: 2500 |
> | --- | --- | --- | --- | --- |
> | FreeMatch | 4.9 | 4.88 | 37.98 | 26.47 |
> | FreeMatch + SEAL | **4.8** | **4.71** | **37.12** | **26** |
>
>
>
> FreeMatch: Self-adaptive Thresholding for Semi-supervised Learning, ICLR 2023

---

> ### Author Response · Authors · 2023-11-23
> **Seeking Your Input on Revised Paper's Alignment with ICLR Standards**
>
> Dear Reviewer DPWp,
>
> As the discussion period approaches its conclusion, **we want to ensure that we have thoroughly addressed all your concerns and that our revised paper fully meets the standards of ICLR**. We would highly value any additional feedback you may provide.
>
> Thank you sincerely for your time and consideration.
>
> Best regards,
>
> The Authors

---

### Official Review · Reviewer_wvK6 · 2023-10-30

**Soundness:** 2 fair
**Presentation:** 1 poor
**Contribution:** 2 fair
**Rating:** 5
**Confidence:** 2

**Summary:**

This paper shows a new mehod that aims to learn the label hierarchy by augmenting the observed labels with latent labels that follow a prior hierarchical structure. Specifically, the authors expand the label alphabet by adding unobserved latent labels to the observed label alphabet.
The data-driven label hierarchy is modeled by combining the predefined hierarchical structure of latent labels with the optimizable assignment between observed and latent labels. In order to verify its rationality, the authors conduct a series of experiments on semi-supervised classification.

**Strengths:**

+ Hierarchical classification and learning class hierarchy via machine learning methods are interesting topics, and this paper bases on a good motivation.
+ This paper provides a good exploration for hierarchical semi-supervised classification tasks, and the experiments can support the idea of this paper to some extent.

**Weaknesses:**

+ In Section 3.2, the authors say that, according to Example 1, the latent variables can provide more information to the classifier and lead to better performance. This confuses me how the conditional probabilities of latent nodes benefit the learning of leaf nodes. In my view, Example 1 only show how to get the marginal probabilities of latent nodes from some given conditional distribution of leaf nodes through Bayes rule.

+ In Eq. (2), the dimension of A is $N$, so the dimension of the latent labels is $N-K$. Does this mean that we need to specify the number of categories of hidden labels in advance? How to set this hyper-parameter for different datasets, and do different values of this parameter affect the performance for one single dataset? Besides, how do we know the number of levels of label hierarchies? In other words, could the proposed method learn a multi-level class hierarchy? This is unclear in the manuscript.

+ In Theorem 2, is $\mathbf 1_N$ a N-dim vector whose elements are all 1? The authors should provide detailed notation for readers.

+ It’s hard to understand Definition 1 since there are so many unclear descriptions. For example, what’s the meaning of $\mu_s$ and $\mu_r$. As mentioned above, $\alpha_{sr}$ is a matrix, but in Eq. (3) it seems a scalar. Those unclear words make make it hard to follow this Section.

+ Is $w_s$ in Eq. (4) a hyper-parameter or a learnable parameters?

+ In Section 3.3, does the Task (a) says that one can pre-define $\mathbf A_1$ rather than learn it by the proposed method? If so, how to perform it? The sentence “we choose A1 to be a trivial binary tree or trees derived from prior knowledge such as a part of a knowledge graph or a decision tree.” is quite unclear.

+ The authors should compare the proposed method with other methods for learning label hierarchy in order to demonstrate its superiority.

**Questions:**

Please refer to the weakness section.

---

> ### Author Response · Authors · 2023-11-22
> **Response to reviewer wvK6**
>
> >Q1: In Section 3.2, the authors say that, according to Example 1, the latent variables can provide more information to the classifier and lead to better performance. This confuses me how the conditional probabilities of latent nodes benefit the learning of leaf nodes. In my view, Example 1 only show how to get the marginal probabilities of latent nodes from some given conditional distribution of leaf nodes through Bayes rule.
>
> A1: Thank you for your question. For example, when you are an image of a red apple and a given connection as depicted in Example 1. As the connection of latent and observed labels are given, one can calculate the ground-truth probability of an object belongs to a latent class. **To accurately predict the image, the prediction on the probability of image belong to latent label, for example color, should also aligns with the previously calculated ground-truth probability.** This gives rise to additional information that may be beneficial to the model.
>
>
> >Q2: In Eq. (2), the dimension of A is $N$, so the dimension of the latent labels is $N-K$. Does this mean that we need to specify the number of categories of hidden labels in advance? How to set this hyper-parameter for different datasets, and do different values of this parameter affect the performance for one single dataset? Besides, how do we know the number of levels of label hierarchies? In other words, could the proposed method learn a multi-level class hierarchy? This is unclear in the manuscript.
>
> A2: Thank you for your question. Yes, you need to specify the number of hidden labels in advance. This hyper-parameter should be set based on the specific dataset used and the prior knowledge. We will conduct ablation studies on the influence of this hyper-parameter and we empirically set this parameter to be close to the number of observed labels. The number of levels can also be seen as a hyper-parameter. As it is determined by $A_1$, it also should be set based on the specific dataset used and the prior knowledge. **Our method can indeed learn a multi-level class hierarchy, please see Figure 2.**
>
>
> >Q3: In Theorem 2, is $1_N$ a N-dim vector whose elements are all 1? The authors should provide detailed notation for readers.
>
> A3: Thank you for your question. Your understanding is correct. We have made this more clear in the revised manuscript.
>
>
> >Q4: It’s hard to understand Definition 1 since there are so many unclear descriptions. For example, what’s the meaning of $\mu_{s}$ and $\mu_{\tau}$. As mentioned above, $\alpha_{sr}$ is a matrix, but in Eq. (3) it seems a scalar. Those unclear words make make it hard to follow this Section.
>
> A4: Thank you for your question. $\mu$ a probability distribution on the total labels. We have added its definition more clearly in the revised version of the manuscript. $\mu_{s}$ and $\mu_{r}$ are the probability value of $\mu$ on coordinate $r$ and $s$. We have defined $\alpha=[\alpha_{sr}]_{s,r}$ in the revised manuscript. Thank you again for your help!
>
> >Q5: Is $w_{s}$ in Eq. (4) a hyper-parameter or a learnable parameters?
>
> A5: Thank you for your question. In this paper, we set it as a fixed hyper-parameter, where all elements of $w$ is 1. **The reason is that we do not want to incorporate additional prior on the weight of edges.**
>
> >Q6: In Section 3.3, does the Task (a) says that one can pre-define $A_1$ rather than learn it by the proposed method? If so, how to perform it? The sentence “we choose A1 to be a trivial binary tree or trees derived from prior knowledge such as a part of a knowledge graph or a decision tree.” is quite unclear.
>
> A6: Thank you for your question. Yes, you need to pre-define a $A_1$. If one does not have much prior knowledge, one may just initialize a random tree's adjacency matrix as $A_1$. If one has more prior knowledge, one can initialize one based on their prior knowledge. We have discussed the details of different tree structures in Appendix A.2.1. The ablation there includes a random initialized one and a tree constructed from prior knowledge.
>
> >Q7: The authors should compare the proposed method with other methods for learning label hierarchy in order to demonstrate its superiority.
>
> A7: Thank you for your question. **As our method adaptively update the hierarchy and can start from random initialized hierarchy, comparing with methods with a good pre-defined hierarchy (which is used in supervised learning approaches, as we discussed in the related work) may be unfair.**  Additionally, we have not come across any other hierarchy exploitation method specifically designed for SSL. Instead, we conducted an ablation study by modifying our prior on tree structures in section A.2.1. It was observed that a more effective prior leads to improved performance.

---

> ### Author Response · Authors · 2023-11-23
> **Seeking Your Input on Revised Paper's Alignment with ICLR Standards**
>
> Dear Reviewer wvK6,
>
> As the discussion period approaches its conclusion, **we want to ensure that we have thoroughly addressed all your concerns and that our revised paper fully meets the standards of ICLR**. We would highly value any additional feedback you may provide.
>
> Thank you sincerely for your time and consideration.
>
> Best regards,
>
> The Authors

---

### Official Review · Reviewer_WAR6 · 2023-11-01

**Soundness:** 2 fair
**Presentation:** 3 good
**Contribution:** 2 fair
**Rating:** 6
**Confidence:** 3

**Summary:**

Semi-supervised learning (SSL) is an effective ML paradigm for learning high-quality classification models by leveraging unlabeled data, especially when annotated data is scarce. A vast number of approaches have been proposed for SSL based on augmentation,  regularization etc. This paper follows the key intuition of enforcing label-label correlations through learned label hierarchies as a an effective regularization strategy to improve SSL performance.

While label hierarchies have been leveraged for SSL earlier, the key novelty of this paper appears to be that it learns a part of the hierarchy by exploiting unlabeled data, thus leading to task-aligned hierarchies. Specifically, the proposed approach learns soft edge weights between a set of latent variables (instantiated to concepts from pre-existing domain-informed hierarchies) and a set of observed variables (actual labels in an SSL). A mathematical framework around Wasserstein distance is proposed for this, although it appears to serve only as an interesting connection and not essential to the central premise of this paper.

Experimental results show significant gains when labeled data is very sparse, mainly when the proposed SEAL regularizer is combined with other previously proposed, effective SSL techniques such as DebiasPL and FlexMatch.

**Strengths:**

* The central idea of simultaneously learning label correlations (via hierarchies) through unlabeled data and then using it to improve SSL performance appears to be novel and useful and deserves more exploration

* Experimental results show significant gains when labeled data is very sparse, mainly when the proposed SEAL regularizer is combined with other previously proposed, effective SSL techniques such as DebiasPL and FlexMatch

**Weaknesses:**

* Experiments do not compare with any competing SSL baselines that leverage label hierarchies. As the related work section notes, several hierarchical schemes have already been proposed, some of them specifically for SSL. But their relative performance w.r.t SEAL has not been compared or discussed. Note that the standalone gains due to SEAL alone are not impressive - only when combined with other orthogonal schemes does it work best. Finally, a good qualitative discussion of results is also lacking. Due to these reasons, the true utility of SEAL has not been demonstrated clearly.

* The set of baselines in Tables 2 and 3 are not the same and no explanation has been provided for this mismatch

* The optimization algorithm for minimizing the final SSL objective (that incorporates SEAL regularizer) has not been fleshed out in full detail



* Theoretical results are not critical. They are more about the sanity of choices made and less about the generalization ability of SEAL technique etc.

**Questions:**

* How does SEAL compare with the best of prior label hierarchy (and label correlation) exploiting approaches for SSL? Does (SEAL+Curriculum) outperform (best label hierarchy baseline for SSL+Curriculum)?

* Why are the set of baselines in Tables 2 and 3 not identical? Please provide full set of results.

* Give more details for (1) hierarchy that is assumed over latent variables, (2) optimization algorithm for minimizing eqn 16.

---

> ### Author Response · Authors · 2023-11-22
> **Response to reviewer WAR6**
>
> Thank you for taking your time to review our paper. We will give detailed responses as follows:
>
> >Q1: Experiments do not compare with any competing SSL baselines that leverage label hierarchies. As the related work section notes, several hierarchical schemes have already been proposed, some of them specifically for SSL. But their relative performance w.r.t SEAL has not been compared or discussed. Note that the standalone gains due to SEAL alone are not impressive - only when combined with other orthogonal schemes does it work best. Finally, a good qualitative discussion of results is also lacking. Due to these reasons, the true utility of SEAL has not been demonstrated clearly.
>
> A1: Thank you for your feedback. **Firstly, in our experiments,  for fair comparison we initialize the tree randomly, which does not incorporate prior knowledge and the works in related work part all used prior knowledge.** Secondly, our method improves the baseline's accuracy with minimal effort and the improvements are significant on STL-10 and CIFAR-10. And we think it is an advantage to consistently improve upon new techniques, which means that SEAL may be incorporated into every new SOTA methods. Lastly, the hierarchy-aware methods in related work are all based on supervised earning. We haven't found any hierarchical-aware method tackles SSL, but we may reproduce some as future work. **Lastly, as our method adaptively update the hierarchy and can start from random initialized hierarchy, comparing with methods with a good pre-defined hierarchy (which is used in supervised learning approaches, as we discussed in the related work) may be unfair.**
>
> >Q2: The set of baselines in Tables 2 and 3 are not the same and no explanation has been provided for this mismatch
>
> A2: Thank you for your inquiry. In our initial paper, we did not include SEAL (Debiased) as a comprehensive comparison due to the absence of CIFAR-100 and STL-10 experiments in the initial paper.
>
> >Q3: The optimization algorithm for minimizing the final SSL objective (that incorporates SEAL regularizer) has not been fleshed out in full detail
>
> A3: Thank you for your feedback. The update formula for SSL is consistent with the formula (16) used in supervised learning. The update rule for the hierarchy is given in Appendix A.6. We first conduct a SGD step, then perform an addition projection operation to project each column of $A_2$ onto the probability simplex.
>
> >Q4: Theoretical results are not critical. They are more about the sanity of choices made and less about the generalization ability of SEAL technique etc.
>
> A4: Thank you for your feedback. As we show in the paper, SEAL can be seen as Wasserstein distance on a posterior tree. **It can have generalization bound given in paper (https://arxiv.org/pdf/2306.04375v1.pdf).**
>
> >Q5: How does SEAL compare with the best of prior label hierarchy (and label correlation) exploiting approaches for SSL? Does (SEAL+Curriculum) outperform (best label hierarchy baseline for SSL+Curriculum)?
>
> A5: Thank you for your question. We do not find any other hierarchy exploiting designed for SSL. **Instead, we do ablation study by changing our prior on tree structures in A.2.1. It is observed that better prior will indeed result in better performance.**
>
> >Q6: Why are the set of baselines in Tables 2 and 3 not identical? Please provide full set of results.
>
> A6: Thank you for your question. We did not include SEAL (Debiased) as a full comparison because the original paper did not conduct experiments on CIFAR-100 and STL-10 datasets. **Furthermore, the main point is to demonstrate that SEAL can be effectively combined with advanced techniques to enhance performance further.**
>
> >Q7: Give more details for (1) hierarchy that is assumed over latent variables, (2) optimization algorithm for minimizing eqn 16.
>
> A7: Thank you for your question. (1) We have discussed the details of different tree structures in Appendix A.2.1. The random hierarchy we initialized in the experiment is a depth-4 tree with 21 nodes, where the latent nodes are organized as depicted in Figure 2 and the other nodes are connected following a random initialization.  (2) The update rule is given in Appendix A.6. We first conduct a SGD step, then perform an addition projection operation to project each column of $A_2$ onto the probability simplex.

---

> > ### Comment · Reviewer_WAR6 · 2023-11-22
> >
> > I appreciate the authors for their clarifications. I would like to keep my score unchanged.

---

### Official Review · Reviewer_LQMB · 2023-11-06

**Soundness:** 3 good
**Presentation:** 3 good
**Contribution:** 2 fair
**Rating:** 5
**Confidence:** 3

**Summary:**

The paper presents a method of constructing label hierarchy using the soft Tree-Wasserstein distance approach. The label hierarchy term is applied as a regularizer for supervised and semi-supervised learning scenarios, where the learning and tree parameters are optimized jointly. The method is compared with various semi-supervised learning baselines and demonstrates accuracy gains when combined with pseudo-labeling techniques.

**Strengths:**

Learning label hierarchy is an important problem especially in the semi-supervised scenarios, where some form of hidden structure is needed to leverage the unlabeled part of a dataset.
When combined with other methods for pseudo-labeling, the method demonstrates its value over several datasets in the semi-supervised setting. The paper is clear and easy to follow, it contains several important ablation studies which illustrate the algorithm behavior.

**Weaknesses:**

The originality of the method is limited. The methodological part of the label hierarchy construction is taken from (Takezawa et al., 2021), and applied as a regularization term to a supervised and a semi-supervised loss.

For the experimental part, it would be great to see the comparison on some other datasets like SVHN and ImageNet, since those are used in the FlexMatch paper, and comparing with FlexMatch is the most important.

For tables 1 and 2, it would be worth adding both SEAL (Debiased) and SEAL (Cirriculum) to both tables to see the full comparison of the proposed scenarios.

**Questions:**

One thing I wasn’t able to get from the experimental study – how do the results differ from different tree structures, depending on the depth and vertex degree of a tree? Some ablation study on that would help.

---

> ### Author Response · Authors · 2023-11-22
> **Response to reviewer LQMB**
>
> Thank you for taking your time to review our paper. We will give detailed responses as follows:
>
> >Q1: The originality of the method is limited. The methodological part of the label hierarchy construction is taken from (Takezawa et al., 2021), and applied as a regularization term to a supervised and a semi-supervised loss.
>
> A1: We appreciate the reviewer's comment regarding the methodological aspect of our work. While it is true that we have incorporated the label hierarchy construction technique described in (Takezawa et al., 2021), we would like to emphasize that though the construction techniques may be similar, our derivation are based on exploiting the latent hierarchical label structure during training and the point of focus is completely different from (Takezawa et al., 2021). **We also point this out in our initial manuscript.** Also the incorporation as regularizer into the loss function allows us to refine the label hierarchy while simultaneously optimizing the model using a combination of labeled and unlabeled data. Hence, our method represents a novel label hierarchy construction technique in the context of supervised and semi-supervised learning.
>
> >Q2: For the experimental part, it would be great to see the comparison on some other datasets like SVHN and ImageNet, since those are used in the FlexMatch paper, and comparing with FlexMatch is the most important.
>
> A2: Thank you for your feedback. **The reason why we did not conduct experiments on SVHN is because the initial accuracy is too high and may be hard to fully convey the effectiveness of our approach.** Due to the computational resource budget and long running time of semi-supervised learning on ImageNet, we will do it as future work.
>
> >Q3: For tables 1 and 2, it would be worth adding both SEAL (Debiased) and SEAL (Cirriculum) to both tables to see the full comparison of the proposed scenarios.
>
> A3: Thank you for your question. We do not add SEAL (Debiased) as full comparison as the initial paper do not conduct CIFAR-100 and STL-10 experiments. **And the gist of it is only to show that SEAL can be combined with advanced techniques to further boost performance.**
>
> >Q4: One thing I wasn’t able to get from the experimental study – how do the results differ from different tree structures, depending on the depth and vertex degree of a tree? Some ablation study on that would help.
>
> A4: Thank you for your feedback. We have done an ablation study in Appendix A.2.1 about different tree structures. We observe that when you do not have a strong prior in the label hierarchy, it is better to incorporate a random tree having moderate depth and vertex degree compared to the number of (observed) labels.

---

> ### Author Response · Authors · 2023-11-23
> **Seeking Your Input on Revised Paper's Alignment with ICLR Standards**
>
> Dear Reviewer LQMB,
>
> As the discussion period approaches its conclusion, **we want to ensure that we have thoroughly addressed all your concerns and that our revised paper fully meets the standards of ICLR**. We would highly value any additional feedback you may provide.
>
> Thank you sincerely for your time and consideration.
>
> Best regards,
>
> The Authors

---

### Author Response · Authors · 2023-11-22
**General Response**

We would like to extend our heartfelt gratitude for the reviewers and ACs dedicated efforts and hard work in assisting with the review process of our manuscript. Your expertise, valuable insights, and constructive feedback have been instrumental in shaping and improving our work.

Based on the reviewer's feedback, we have added more precise definitions to some terms used in the paper.

---

### Meta-Review · Area_Chair_reba · 2023-12-07

**Metareview:**

This paper proposes a framework to jointly learn and exploit label hierarchy to boost the performance. The label hierarchy term is applied as a regularizer for both supervised and semi-supervised learning scenarios, where the learning and tree parameters are optimized jointly. In experiments, the method is compared with various semi-supervised learning baselines and demonstrates accuracy gains when combined with pseudo-labeling techniques.

The key idea of this paper is the data-driven label hierarchical learning part, which is different from previous predefined label correlation exploitation. However, there are several concerns according to the originality, theoretical analysis and experimental evaluation. Based on the overall reviews, I am inclined to reject this paper.

**Justification For Why Not Higher Score:**

1. The technical contribution is not enough. As mentioned by Reviewer LQMB, the methodological part of the label hierarchy construction is taken from (Takezawa et al., 2021), and applied as a regularization term to a supervised and a semi-supervised loss.

2. Theoretical results are not critical. As mentioned by Reviewer DPWp, the theoretical analysis presented lacks valuable insights for the proposed approach. Furthermore, the author fails to clearly and concisely describe the integration of theory with their proposed approach, as mentioned in the Weaknesses 1;

3. All reviewers agree that the experiments are not convincing enough to demonstrate the effectiveness of the proposed approach.

**Justification For Why Not Lower Score:**

N/A.

---

### Decision · Program_Chairs · 2024-01-16

Reject